# A Review of DUSP26: Structure, Regulation and Relevance in Human Disease

**DOI:** 10.3390/ijms22020776

**Published:** 2021-01-14

**Authors:** Elliott M. Thompson, Andrew W. Stoker

**Affiliations:** Developmental Biology & Cancer Department, UCL Great Ormond Street Institute of Child Health, 30 Guilford Street, London WC1N 1EH, UK; elliott.m.thompson18@ucl.ac.uk

**Keywords:** dual specificity phosphatase, MAP kinase phosphatase, cancer, neuroblastoma, phosphatase inhibitor

## Abstract

Dual specificity phosphatases (DUSPs) play a crucial role in the regulation of intracellular signalling pathways, which in turn influence a broad range of physiological processes. DUSP malfunction is increasingly observed in a broad range of human diseases due to deregulation of key pathways, most notably the MAP kinase (MAPK) cascades. Dual specificity phosphatase 26 (DUSP26) is an atypical DUSP with a range of physiological substrates including the MAPKs. The residues that govern DUSP26 substrate specificity are yet to be determined; however, recent evidence suggests that interactions with a binding partner may be required for DUSP26 catalytic activity. DUSP26 is heavily implicated in cancer where, akin to other DUSPs, it displays both tumour-suppressive and -promoting properties, depending on the context. Here we review DUSP26 by evaluating its transcriptional patterns, protein crystallographic structure and substrate binding, as well as its physiological role(s) and binding partners, its role in human disease and the development of DUSP26 inhibitors.

## 1. Introduction

### 1.1. Protein Tyrosine Phosphatases

Protein phosphorylation is a post-translational modification that is widely regarded as a global regulatory mechanism for modulating protein function. Consequently, during a cellular response to internal or external stimuli, alterations in protein phosphorylation can dictate the downstream signalling pathways. Protein tyrosine phosphatases (PTPs) and receptor tyrosine kinases (RTKs) regulate the phosphorylation of tyrosine residues. Historically, RTKs have been researched more thoroughly since they were believed to drive intracellular signalling and PTPs were considered more as phosphate scavengers with no regulatory role [1]. However, PTPs are now established regulators of signalling in their own right and as PTP malfunction has been observed in a range of human diseases these enzymes are potential therapeutic targets [2,3]. Over 100 PTP superfamily proteins have been identified. With the exception of a handful of pseudophosphatase members, all have an active site cysteine that is critical for dephosphorylation activity. These proteins can be divided into four classes. Classes 1–3 contain a Cys-based active site whereas class 4 contains an Asp-based active site [3]. Class 1 constitutes the majority of these genes (99/107) and can be further sub-divided into the ‘classical’ PTPs and ‘VH1-like’ dual-specificity phosphatases (DUSPs). PTPs are highly modular as ~75% contain an additional domain or motif on top of the core catalytic domain providing more diverse functions and regulation [3]. Further regulation can also be achieved transcriptionally through alternative promoters, post-transcriptionally through splicing variants and post-translationally through binding partners, dimerisation and post-translational modifications [2,4].

### 1.2. DUSPs

Unlike their classical cousins, DUSP enzymes can dephosphorylate both tyrosine and serine/threonine residues plus other targets such as lipids. The first DUSP identified was VH1 from the vaccina virus [5]. Since then, 61 human DUSP genes have been identified, broadly divided into seven classes: the MAP-kinase phosphatases (MKPs), slingshot phosphatases (SSHs), atypical DUSPs (aDUSPs), PRLs, CDC14s, PTENs and myotubularins [6]. Interestingly, not all DUSPs dephosphorylate amino acids. The physiological substrates of PTEN and myotubularins are inositol phospholipids. In addition, certain aDUSPs can dephosphorylate glucans and the 5′ CAP structure of mRNA (Table 1) [7,8]. Nonetheless, all DUSPs demonstrate a similar catalytic mechanism. Finally, each member of the DUSP family has various synonyms, including, confusingly, names of other DUSPs in the same subgroup. Thus, to avoid confusion we will use their gene names as documented in the HUGO gene nomenclature committee (HGNC).

### 1.3. Atypical DUSPs

The aDUSPs are a heterogenous group of 20 DUSPs with highly conserved active site motifs and the ability to dephosphorylate a range of different substrates (Table 1) [8]. The aDUSPs have been compared to the MKPs due to their analogous catalytic domains and HCX5R motif and have therefore been studied predominantly in terms of MAP kinase (MAPK) regulation (Figure 1B) [6,22]. However, aDUSPs lack the MAPK-binding domain (MBD) that governs MKPs substrate specificity. The absence of the MBD in the aDUSPs suggests that (i) they can dephosphorylate non-MAPK substrates and (ii) alternative regulatory mechanisms are in place to regulate their function. This is highlighted by DUSP3, the most researched aDUSP (recently reviewed by [9,23,24]). Although DUSP3 is able to dephosphorylate ERK, JNK and p38, it can also dephosphorylate tyrosine residues of non-MAPK substrates such as STAT5, EGFR and FAK [9]. Moreover, where the MKPs are upregulated transcriptionally by MAPK signalling and translationally by MAPK binding, DUSP3 is unaffected by MAPK signalling and its catalytic activity is increased by binding to a range of non-MAPK proteins [9,25].

A second aDUSP that has seen a great deal of interest, and being the main subject of this review, is DUSP26. Initial literature on DUSP26 suggested that it could dephosphorylate p38 and upregulate JNK and ERK [26,27]. However, as with DUSP3, emerging substrates and binding partners of DUSP26 have been identified which link it to the progression of several human diseases. For example, DUSP26 is implicated in cancer, with potential roles in neuroblastoma, anaplastic thyroid cancer, glioblastoma and breast cancer, among others [19,28,29,30,31]. Inhibition of specific PTP family members, either experimentally or for potential therapeutic goals, is notoriously difficult. However, various small molecule inhibitors have been generated against DUSP26 [32,33,34]. Although they do not yet display the potency or selectivity to be used clinically, their scaffolds serve as a good starting point for the generation of more specific inhibitors. The aim of this review is to summarise the current knowledge on DUSP26, looking at its transcriptional patterns, protein crystallographic structure and substrate binding, as well as its physiological role(s) and binding partners, its role in human disease and the development of DUSP26 inhibitors.

## 2. *DUSP26* Gene and Protein Structure

### 2.1. Gene Identification, Expression and Protein Localisation

The *DUSP26* gene was identified independently by three research groups who each assigned the encoded protein a different name: mitogen-activated protein kinase phosphatase-8; low molecular-mass DUSP-4; and neuroendocrine-associated phosphatase, although the official HGNC gene name is now *DUSP26* [14,15,35]. The *DUSP26* gene is located on chromosome 8p12, an unstable region often lost or gained/amplified in cancer (discussed below). It is worth noting that *DUSP26* contains two alternative start codons at Met11 and Met14, but it remains unclear whether either of these sites are preferentially used compared to the first start codon [19].

The DUSP26 protein sequence is homologous to DUSP3, DUSP13A, DUSP13B, DUSP29 and STYXL2, especially surrounding the active site region, as they share a high degree of sequence similarity and appear under the same clade of the DUSP phylogenetic tree (Figure 1A,C). Compared to the MKPs, the aDUSPs display less sequence conservation, highlighted by their lower bootstrap confidence levels, which is unsurprising considering their broader range of substrates and physiological functions. Initial studies demonstrated that *DUSP26* mRNA expression is low in most normal human and mouse tissues except the brain, heart and skeletal muscle [14,15,36]. Interestingly, DUSP13A, DUSP13B and DUSP29 also display a similar degree of tissue-restricted expression [35,36]. As all are still expressed in skeletal muscle, this suggests that their common ancestor may have originally been restricted to skeletal muscle before diverging into other tissues. In the cell, DUSP26 is located predominantly in the nucleus, although weak staining has been observed in the cytoplasm, whereas DUSP13A and DUSP13B and DUSP29 are cytosolic proteins [26,27]. DUSP3 also localises to the nucleus and cytoplasm and, unlike most other aDUSPs, both DUSP3 and DUSP26 have a broad range of MAPK and non-MAPK substrates [9]. Their sequence similarity and broad functions might suggest a role of DUSP3 and DUSP26 in overlapping physiological processes.

### 2.2. Protein Structure

#### 2.2.1. General Structural Considerations

The DUSP26 protein is 211 amino acids in length and contains a non-catalytic N-terminal domain (1–60; NTD) and a catalytic C-terminal domain (61–211; CTD) (Figure 2A). The CTD contains the aforementioned HCX5R motif and an AYLM motif which is conserved among MKPs (Figure 1B) [27]. Initial crystallographic studies of the DUSP26 CTD (DUSP26-C) deemed it catalytically inactive [40,41]. The PTP-loop was positioned in a distorted orientation, which disfavored substrate entry and is likely to disrupt key interactions such as those described recently in the DPN-loop in the related DUSP22 enzyme [42]. However, a subsequent structure, containing an N-terminal extension (39–211; DUSP26-N) demonstrated catalytic activity [43]. Although the overall structure of DUSP26-N was similar to DUSP26-C, the C-terminal α7–α8 loop stabilised residues 155–157, creating an active PTP-loop conformation (Figure 2B). This study highlighted a crucial scaffolding role of α1, not present in DUSP26-C, in positioning the α7–α8 in the optimal orientation for contacting the PTP-loop residues.

#### 2.2.2. Active Site

In accord with other DUSPs, DUSP26 contains the characteristic catalytic triad composed of a Cys152, Arg158 and Asp120 (Figure 1C and Figure 2B) [43]. The catalytic mechanism of PTPs is well established, proceeding via a sequential, 2-step dephosphorylation mechanism using acid/base catalysis [44]. As mentioned previously, the DUSP26 PTP-loop forms a canonically active conformation highly resembling that of DUSP3 and DUSP29 (Figure 2C). The PTP-loop backbone amide bonds in DUSP26-N, DUSP3 and DUSP29 point inwards, allowing hydrogen bond formation that stabilises substrate entry. Furthermore, these residues are critical in generating the positive electrostatic potential which promotes nucleophilic attack by the catalytic cysteine [43]. In contrast, the DUSP26-C PTP-loop amide bonds are distorted, pointing away from the active site, demonstrating why it is inactive in vitro (Figure 2C). Mutagenesis experiments have also revealed that Arg186 is important to DUSP26 phosphatase activity because its mutation to a leucine or glutamine abrogated hydrogen bonding with Gly155 and decreased in vitro catalytic activity significantly (Figure 2A) [33]. The same study also took advantage of the similarity between DUSP26 and the DUSP13B homolog to generate a DUSP26 pharmacophore model for the generation of DUSP26 chemical inhibitors (discussed below).

#### 2.2.3. Protein Dimerisation

Dimerisation that can positively or negatively impact function has become recognised as an important regulatory mechanism among DUSPs [45,46,47,48,49]. Both DUSP3 and DUSP29 homodimerise in solution via N-terminal domain swapped dimers [48,49]. In DUSP3, homodimerisation is governed by residues Phe68 and Met69 in the α4-β3 loop and is a regulatory mechanism to suppress catalytic activity [48]. The result of DUSP29 dimerisation on catalytic activity is unknown. Although DUSP26-N is monomeric in solution, DUSP26-C exists as a mixture of monomer and dimer. When superimposed to DUSP3 and DUSP29, DUSP26 has very similar overall folds; however, there is a crucial deviation in the α4-β3 loop, critical for DUSP3 homodimerisation (Figure 2D,E). Additionally, there are major differences in terms of their structural topology and electrostatic interactions [43]. Given its global structural similarity to DUSP3 and DUSP29 and its lack of regulatory regions, it is plausible that interactions between DUSP26 and a binding partner alter its conformation, particularly that of the α4-β3 loop, permitting dimerisation and regulating function in vivo [40]. Further studies would be required to examine this.

## 3. Regulation and Binding Partners

### 3.1. Transcriptional Regulation

The restricted expression of DUSP26 protein in certain tissues suggests that DUSP26 has specific physiological functions that are most likely intricately regulated [14,15,36]. To date, regulation of DUSP26 protein level is thought to occur predominantly at a transcriptional level and also interactions with binding proteins post-translationally (Figure 3). Patterson et al. demonstrated that *DUSP26* gene expression is downregulated epigenetically by both histone de-acetylation and promoter methylation [50]. Treatment with trichostatin A, a histone de-acetylase inhibitor, increased *DUSP26* mRNA in ovarian cancer cell lines. *DUSP26* contains a 600 bp CpG island situated 100bp of the transcriptional start site. Hypermethylation of this was observed in ovarian, neuroblastoma and brain cancer cell lines and is part of a CpG-based signature for lung adenocarcinoma and colorectal cancer [50,51,52,53]. Genome-wide single-nucleotide methylation profiling in HEK293 cells revealed that the *DUSP26* promoter was methylated by TET dioxygenases [54]. An in-depth analysis into *DUSP26* hypermethylation across different tissues would be beneficial in understanding the physiological cues governing *DUSP26* silencing.

### 3.2. Post-Translational Regulation: Binding Partners

After the discovery of DUSP26, an array of in vivo studies provided conflicting evidence as to DUSP26 MAPK substrate specificity. In brief, independent experiments demonstrated that DUSP26 either dephosphorylated p38 and ERK, activated JNK and p38, or had no effect on the MAPKs [14,15,23,36,52]. As well as the different systems used, this suggests that alternative regulatory mechanisms, depending on the cellular context, modulate DUSP26 activity. There is indeed accumulating evidence of novel DUSP26 binding partners that regulate DUSP26 function in vivo, potentially causing these conflicting results. Such interactions are discussed below and are presented in Figure 3.

#### 3.2.1. HSF4b

The first DUSP26 interacting protein identified was Heat shock transcription factor 4 isoform B (HSF4b). The yeast-2-hybrid system and co-immunoprecipitation both confirmed that DUSP26 interacts with HSF4b [55]. In HI1299 cells, HSF4b binds ERK and DUSP26 simultaneously and, in doing so, DUSP26 dephosphorylates ERK and inhibits ERK-mediated phosphorylation of HSF4b. DUSP26 therefore indirectly regulates HSF4b through ERK inactivation but, critically, requires binding to HSF4b to function. A bioinformatic analysis of the HSF4b interaction network reinforced this, with HSF4b demonstrating the largest physical interaction with DUSP26 [56]. High HSF4b expression is an indicator of poor survival in colorectal cancer (CRC). As the *DUSP26* CpG island is hypermethylated in CRC, further studies might reveal that DUSP26-HSF4b interactions are tumour-suppressive in this setting (Table 2) [52,53].

#### 3.2.2. SCRIB

SCRIB is an adaptor protein involved in regulating cell polarity [14] and it binds and dephosphorylates ERK by recruiting Protein Phosphatase 1γ [68]. Sacco et al. identified a putative PDZ-domain binding motif in the DUSP26 CTD which binds the 4th SCRIB PDZ domain [14]. As DUSP26 siRNA increased ERK phosphorylation in HeLa cells [69], the study speculates that SCRIB acts as a molecular bridge allowing DUSP26-mediated dephosphorylation of ERK, although other molecular substrates are plausible (Figure 4).

#### 3.2.3. KIF3

In IMR32 neuroblastoma cells, DUSP26 co-immunoprecipitated with N-cadherin, β-catenin and the KIF3 motor complex via the KIF3a subunit [19]. DUSP26 was also able to dephosphorylate a second KIF3 motor complex subunit, KAP3 and the DUSP26 NTD was essential to both binding KIF3a and dephosphorylating KAP3. The KIF3 motor complex is part of a large superfamily of kinesin motors that transport vesicles and proteins via microtubule filaments [70]. Interestingly, a recent report demonstrates that phosphorylation of KAP3 by the MARK2 phosphorylation cascade can suppress KIF3 motor complex function [71]. As subcellular transport of β-catenin/N-cadherin is mediated through the KIF3 motor complex, DUSP26 activity may underpin transport of β-catenin/N-cadherin to axon ends, through dephosphorylation of KAP3. In line with this, DUSP26 expression enhances N-cadherin-mediated cell–cell adhesion and promoted N-cadherin and β-catenin localisation at cell–cell contact sites [19]. Curiously, as this was independent of the DUSP26 NTD, it is unclear how DUSP26 governs N-cadherin and β-catenin localisation, although it is likely to be through regulation of the KIF3 motor complex. Moreover, β-catenin knockout mice display significantly reduced DUSP26 expression [72]. Taken together, these findings suggest that there may be a positive feedback mechanism between β-catenin and DUSP26 whereby β-catenin upregulates *DUSP26* gene expression and DUSP26 in turn promotes β-catenin localisation at cell–cell contact sites via upregulation of the KIF3 motor complex subunit.

#### 3.2.4. AK2

The phosphotransferase enzyme AK2 also interacts with DUSP26, in this case regulating FADD phosphorylation. In HeLa cells, AK2 directly interacted with FADD, but FADD phosphorylation was independent of AK2 catalytic activity [61]. Kim et al. demonstrated that DUSP26 co-immunoprecipitated with AK2 and dephosphorylated FADD at Ser194 [61]. Crucially, recombinant AK2 increased DUSP26 phosphatase activity against p-FADD in vitro and in HEK293 cells. This suggests that AK2 binding to FADD and DUSP26 is required for DUSP26 to make optimal contacts with FADD, in line with the proposed mechanism presented in Figure 4.

#### 3.2.5. TAK1

DUSP26 is also regulated by direct binding to TAK1, a regulator of NF-*κ*B and MAPK signalling [18,73]. In HEK293 cells, DUSP26 directly interacted with TAK1 via its NTD [18]. Subsequently in LO2 cells, DUSP26 overexpression attenuated TAK1, JNK, and p38 phosphorylation. However, co-transfection of constitutively active (CA) TAK1 abrogated this effect. Therefore, TAK1 signalling through JNK and p38 was regulated by DUSP26 binding but this could be overcome by overexpression of CA-TAK1.

The studies presented above suggest that DUSP26 often requires a scaffolding protein to act as a molecular bridge. This would bring DUSP26 and the substrate into close proximity, possibly governing substrate specificity and increasing the dephosphorylation rate of the target (Figure 4) [14]. As AK2 enhanced DUSP26-mediated dephosphorylation of FADD but was not required, it is plausible that the binding partner can alter DUSP26 conformation to promote phosphatase activity. There are hints that DUSP26 is regulated in this manner by a range of other proteins. Two large proteomic screens in Hela and INS-1 β-cells have been performed to identify novel DUSP26 binding partners [14,74]. Subsequent KEGG pathway enrichment analysis identified binding partners involved in cell death and DNA-damage response, suggesting that DUSP26 functions in cell survival. Future experiments validating these interactions will shed light onto the mechanism of DUSP26 catalytic activity and the potential role of DUSP26 in cell survival.

## 4. Function in Relation to Human Disease

### 4.1. Cancer Roles of DUSP26

DUSP26 displays both tumour-suppressive and -promoting properties in different contexts. This is in accord with many other DUSPs, especially MKPs, which can also be pro- or anti-tumour survival [12,66,67]. The potential role of DUSP26 in each cancer type is discussed below and summarised in Table 2.

#### 4.1.1. Neuroblastoma

The role of DUSP26 in neuroblastoma (NB) is controversial. One study demonstrated that DUSP26 mRNA was downregulated in NB cell lines compared to normal adrenal tissue, implicating DUSP26 as a potential tumour suppressor [50]. In accordance, low *DUSP26* mRNA expression is highly associated with decreased patient survival in NB primary samples [75] (Figure 5). However, this contradicts a report demonstrating DUSP26 overexpression in high-risk NB compared to the adrenal gland [59]. Additionally, the bulk of the direct experimental literature supports growth-promoting roles of DUSP26, for example with DUSP26 knockdown decreasing cell proliferation in both NB cell lines and a xenograft model [30]. In NB cell lines, exogenous DUSP26 expression overcame Doxorubicin-induced apoptosis through dephosphorylation of p53. Interestingly, in vitro DUSP26 physically interacted with and dephosphorylated p53 [59]. P53 contains a range of phospho-acceptor sites, many of which are essential to p53 activation and function [76,77]. Treatment with NSC-87877, a DUSP26 inhibitor, led to activation of p38, p53, Caspase 3 and increased cleaved PARP, demonstrating the tumour-promoting roles of DUSP26 [30]. The researchers conclude that the oncogenic role of DUSP26 is mediated through dephosphorylation of p53 directly at Ser37 and indirectly via p38. In another study, DUSP26 knockdown in INS-1E cells led to upregulation of p53 phosphorylation at Ser15, further suggesting that DUSP26 antagonises p53 activity [78]. These data are in accord with the previously mentioned proteomic screens that identified DUSP26 binding partners involved in the DNA-damage response and cell survival [14,74]. Although p53 was not identified as a direct substrate in these screens, they demonstrate very similar physiological roles of DUSP26 in DNA damage and cell stress.

N-Myc is overexpressed in approximately 25% of patients and is a routine biomarker for NB risk stratification [79]. The N-Myc protein is a very broad transcriptional regulator and Chip-Seq analysis identified the *DUSP26* gene region as an N-Myc target which is upregulated in MYCN-amplified NB cells [80]. This suggests that the *DUSP26* gene is regulated by N-Myc and this may influence N-Myc’s tumorigenic potential. This study only used one cell line, therefore studies in further lines are needed for validation of these results and to identify the downstream consequence of *DUSP26* upregulation in MYCN-amplified NB.

It is well known that NB is ameliorated by inducing differentiation, highlighted by the use of retinoid-based treatments in NB [81,82,83]. In J1 mouse ESCs, retinoic acid-induced neuronal differentiation was accompanied by significant upregulation of *DUSP26* mRNA [84]. Similarly, in PC12 cells, NGF treatment induced cell differentiation and increased *DUSP26* expression [85]. Paradoxically, exogenous expression of DUSP26 prior to NGF treatment suppressed NGF-induced neuronal differentiation. As this suppression was not observed by catalytically inactive DUSP26, this suggests that DUSP26 actively modulates neuronal cell differentiation [85]. Interestingly, DUSP26 prevented NGF-induced differentiation of PC12 cells through direct and indirect downregulation of RTKs. First, DUSP26 overexpression led to dephosphorylation of TrkA and FGFR1, two RTKs involved in neuronal differentiation and NGF signalling [31,79,80]. DUSP26 could dephosphorylate affinity-purified TrkA and FGFR1 and this was enhanced by the addition of AK2, a known binding partner of DUSP26; however, no native AK2-DUSP26 complexes were identified in PC12 cells [17]. Second, DUSP26 also inhibits EGFR, another RTK that promotes NGF-induced differentiation [62]. Exogenous DUSP26 expression suppressed EGFR promoter activity, protein expression and downstream signalling through AKT, ERK1/2 and ERK5. Although DUSP26 mediated EGFR downregulation via inhibition of AKT/PI3K, this is most likely indirect and potentially due to dephosphorylation of TrkA and FGFR1 [62]. Taken together, although clinical data suggest a tumour-suppressive role of DUSP26, the majority of the experimental data instead suggest that DUSP26 antagonises p53-mediated cell death and promotes the dedifferentiated state typical of aggressive NB.

#### 4.1.2. Anaplastic Thyroid Cancer

The 8p region containing the *DUSP26* gene is often a target of amplification or loss of heterozygosity in cancer [17,20,82,83]. A recent study identified a range of breakpoints at 8p11-p12 demonstrating the instability in this region [29]. The 8p12 region was amplified in anaplastic thyroid cancer (ATC) primary tumours and cell lines and *DUSP26* mRNA was upregulated in cases exhibiting copy number increases, suggesting a potential role of DUSP26 in promoting tumour cell growth [31]. In ATC cells with low DUSP26 expression, transfection of DUSP26 cDNA promoted cell growth whereas DUSP26 knockdown in cells that had amplified *DUSP26* decreased their growth. DUSP26 is suggested to function here through inhibition of p38-mediated apoptosis. Exogenous DUSP26 expression in ATC cells prevented anisomycin-induced upregulation of phosphorylated p38 and caspase 3, whereas cells expressing the DUSP26 C152S mutant did not. Surprisingly, this is the only study investigating DUSP26 in ATC. Considering the significant effect of DUSP26 knockdown in cells with amplified DUSP26, the therapeutic potential of targeting DUSP26 in this setting is interesting.

#### 4.1.3. Breast Cancer

In breast cancer, both loss of heterozygosity (LOH) and amplification of the 8p12 region have been observed [29,61]. One study analysed 234 breast tumour samples and identified an increase and decrease of *DUSP26* gene copy number in 16% and 10% of cases, respectively [29]. In both events, *DUSP26* gene copy number was significantly associated with overall patient survival. Interestingly, in patients on chemotherapy treatment, a gain of *DUSP26* predicted worse survival whereas on radiotherapy treatment, a loss of *DUSP26* predicted worse survival. Results were validated with a clinical dataset, crucially demonstrating that *DUSP26* gene amplification did indeed result in higher *DUSP26* mRNA expression. In another study, a decrease in *DUSP26* mRNA expression was observed in 8 of 14 primary breast tumour samples [61]. This correlated with loss of the DUSP26 binding protein AK2 and an increase in FADD phosphorylation. It would be intriguing to see whether the LOH of 8p12 breast cancer specimens correlates with an increased FADD-Pi which would support the functional effect of loss of *DUSP26* in breast cancer.

#### 4.1.4. Glioblastoma

A tumour-suppressive role of DUSP26 is also seen in glioblastoma. Initial studies demonstrated that *DUSP26* expression was downregulated in 7/9 primary glioma specimens compared to normal brain [19]. These findings were corroborated by Bourgonje et al. who investigated the PTPome mRNA expression profile across 83 diffuse glioma samples [28]. *DUSP26* mRNA were significantly reduced in tumour specimens compared to normal brain tissue. Furthermore, the higher the histological grade of the tumour, the lower the DUSP26 protein levels. These findings were validated using a collection of clinical datasets where high-grade tumours demonstrated reduced *DUSP26* mRNA expression and lower survival probability. Supporting the potential suppressive role, DUSP26 overexpression in E98 glioblastoma cells caused a reduction in cell proliferation accompanied by a reduction in spheroid outgrowth [28].

#### 4.1.5. T-Cell Lymphoblastic Lymphoma

As mentioned previously, DUSP26 binding to AK2 allows it to dephosphorylate FADD in HeLa cells [61]. In a set of 23 human T-cell lymphoblastic lymphoma (T-LBL) samples, FADD expression was downregulated in 35% [86]. Marín-Rubio et al. hypothesised that FADD reduction impaired Fas-mediated cell death, promoting tumour cell growth [57]. Although only eight samples were analysed (four healthy and four T-LBL tumours), they demonstrated that DUSP26 protein was upregulated in three T-LBL tumours and none of the healthy controls. Following this, treatment with a DUSP26 inhibitor increased pFADD in BW5147.3 and JURKAT cells, suggesting that DUSP26 functions here as it does in HeLa cells. Further work identifying the AK2-DUSP26 interaction in these cells and whether it drives cell growth is required to confirm the role of DUSP26 in T-LBL.

### 4.2. Non-Cancer Roles of DUSP26

#### 4.2.1. Nonalcoholic Fatty Liver Disease

Nonalcoholic fatty liver disease is a form of chronic liver disease characterised by progressive steatosis and is significantly associated with insulin resistance and obesity [87,88]. As various DUSPs have been implicated in lipid metabolism and hepatocyte cell survival, a recent study investigated the role of DUSP26 using hepatocyte-specific *DUSP26*-knockout (KO) and -transgenic mice [18]. *DUSP26*-KO mice that were fed a high fat diet (HFD) exhibited lipid accumulation and an upregulation and downregulation of the mRNA expression for lipid synthesis and oxidation genes, respectively. DUSP26 inhibited insulin resistance as HFD-fed *DUSP26*-KO mice displayed higher insulin plasma levels and lower glucose tolerance. The reduction in insulin sensitivity correlated with an upregulation of pro-inflammatory cytokines and a decrease in NF-κB signalling. As mentioned previously, DUSP26 dephosphorylates TAK1 to abrogate its intracellular signalling, but this could be overcome by CA-TAK1 [18]. In L02 cells, overexpression of CA-TAK1 also reversed the DUSP26 effects on lipid accumulation, mRNA of lipid synthesis genes and inflammatory cytokines [18]. The rescue effect of TAK1 is yet to be confirmed in transgenic mice, but results suggest that TAK1-activation of NF-kB drives insulin resistance and lipid accumulation and is at least partially controlled by DUSP26.

#### 4.2.2. Diabetes and Diabetic Nephropathy

DUSP26 demonstrated a pro-survival role in pancreatic β-cells [78]. The microRNA-200 (miR-200) family induces β-cell death and downregulates β-cell survival genes [78,89]. In HEK293 cells, miR-200c overexpression significantly reduced DUSP26 mRNA expression. DUSP26 knockdown in INS-1E cells reduced pancreatic β-cell counts and induced apoptosis via p53 signalling. Whether DUSP26 dephosphorylates p53 directly or via an intermediate protein was not investigated. As DUSP26 has no miRNA response element, Belgardt et al. identified various intermediate proteins downregulated by miR-200c which regulated *DUSP26* expression [78].

On a similar note, DUSP26 displayed a pro-survival role in diabetic nephropathy (DN). DN results from a progressive deterioration of the kidney and renal injury in a subset of diabetic patients [90]. Expression of DUSP26 was significantly lower in renal samples from patients with DN compared to healthy tissue [91]. In DN mice induced by streptozotocin (STZ), *DUSP26*-KO increased blood glucose levels and albuminuria, a routine biomarker for DN. Immunohistochemical analysis on the globular basement membrane revealed a decrease in nephrin expression in *DUSP26*-KO mice, indicating that DUSP26 suppresses STZ-induced podocyte injury [91]. As podocyte injury is associated with oxidative stress, the researchers investigated the contribution of DUSP26 to the production of reactive oxygen species (ROS). In both DN mice and differentiated mouse podocytes, loss of DUSP26 increased STZ-induced ROS production which in turn upregulated the MAPK pathways and TGF-β1 signalling. It is worth noting that ROS inhibits PTP activity by oxidising the catalytic cysteine to sulfenic acid, suggesting that loss of DUSP26 in DN mice would be exacerbated by ROS production [92]. Pre-treatment of mouse podocytes with NAC, a ROS scavenger, reversed the upregulation of the MAPK pathways and TGF-β1, suggesting that DUSP26 was antagonising the effects of oxidative stress [91]. Overall, DUSP26 demonstrates a protective role in DN since *DUSP26*-KO exacerbates podocyte injury via oxidative stress, leading to activation of the MAPKs and TGF-β1.

#### 4.2.3. Alzheimer’s Disease

It was reported that *DUSP26* is overexpressed in the hippocampus of Alzheimer’s disease patients [93]. Accumulation of amyloid-β protein (Aβ) is a major hallmark of Alzheimer’s disease and in HEK293 cells, DUSP26 increased γ-secretase-mediated cleavage of the amyloid precursor protein into Aβ [93,94]. Subsequently, in SH-SY5Y cells pre-treated with retinoic acid, DUSP26 directed the γ-secretase complex to axonal regions of neuronal cells. Interestingly, DUSP26 mediated these effects though phosphorylation of JNK since the addition of a JNK inhibitor prevented hypoxia-induced Aβ generation. Positive regulation of JNK and p38 has also been identified previously in sorbitol-induced COS-7 cells where exogenous DUSP26 expression increased basal phosphorylation of JNK and p38 [27]. It is worth noting that similar functions have been observed by DUSP13A, the DUSP26 homolog, and by DUSP23 [95,96]. In line with these findings, these reports suggest that DUSP26 could also function as a scaffold in neuronal cells by recruiting an intermediate kinase that phosphorylates and activates JNK which subsequently aids γ-secretase localisation or function.

## 5. Chemical Inhibition of DUSP26

Historically, there has been a stigma attached to developing inhibitors against PTPs due to the notion that they were ‘undruggable’ targets [97]. The difficulty in generating a clinical inhibitor primarily stems from in vivo target selectivity and cell permeability: PTP enzyme active sites are highly conserved across the family and are highly polar, leading to generally polar inhibitors with poor cell permeability. Consequently, researchers have commonly used broad-specificity PTP inhibitors as tools to characterise PTP loss of function with the hope that they could be developed into clinical therapeutics. An example of this is the widespread, experimental use of oxidovanadium complexes [98], some of which have again shown effective anti-tumour activity recently [99]. Moreover, their recent, successful delivery to cultured tumour cells in liposomes may further potentiate their application in vivo [100]. In cells, oxidovanadium complexes ultimately speciate into vanadate, the reversible inhibitor of PTPs, and although this would inhibit DUSPs it is non-selective across the enzyme family. Therefore, given the growing realisation that aberrant PTP function significantly contributes to disease, extensive research has continued to strive for more selective, cell permeable inhibitors [97,101]. To date, three chemical inhibitors have been identified which are somewhat specific towards DUSP26 (Table 3).

### 5.1. Ethyl-3,4-dephostatin

Ethyl-3,4-dephostatin is a stable analog of the broad-spectrum PTP inhibitor dephostatin. Seo and Cho investigated the effect of Ethyl-3,4-dephostatin on 13 PTPs and demonstrated concentration-dependent phosphatase inhibition of DUSP26 [32]. In vivo, ethyl-3,4-dephostatin is able to inhibit DUSP26-mediated dephosphorylation of p38 and p53. Unfortunately, ethyl-3,4-dephostatin displays low substrate selectivity as it inhibits a range of other PTPs including SHP-1 and PTP-1B (Table 3).

### 5.2. NSC-87877

In 2006, NSC-87877 was discovered as a competitive inhibitor of SHP-2 [106]. Selectivity between SHP-1 and -2 was indistinguishable but there was less potency against five other human PTPs. Following this, Song et al. screened NSC-87877 against 15 PTPs where the highest inhibition was against DUSP26 [34]. It is worth noting that DUSP26 inhibition was 3-fold higher than that of SHP-1. Results were confirmed in HEK293 cells where NSC-87877 inhibited DUSP26-mediated dephosphorylation of p38 [34]. As previously mentioned, *DUSP26* is overexpressed in NB and promotes tumorigenesis in xenografts by dephosphorylating p53 [59]. In NB cell lines, treatment with NSC-87877 reduced cell proliferation [30]. NSC-87877 demonstrated on-target inhibition as SHP-1 shRNA did not affect NB cell growth and NSC-87877-mediated cell death was reversed by knockdown of p53. In vivo, xenografted NB tumours were significantly smaller after treatment with NSC-87877. These data support the therapeutic potential of targeting DUSP26 in NB using NSC-87877. In another study, NSC-87877 inhibited DUSP26-mediated activation of JNK and subsequent accumulation of Aβ [93]. However, in another study NSC-87877 also decreases Aβ production in HEK293 cells, but through inhibition of SHP-2 [106]. This demonstrates the potential of NSC-87877 in Alzheimer’s disease although its physiological target is unclear. It should be acknowledged that various studies have used NSC-87877 as selective inhibitors of SHP-1/2 without considering the potential off-target inhibition of DUSP26, or any other PTP for that matter. As with ethyl-3,4-dephostatin, issues surrounding selectivity remain a recurrent theme, highlighting the importance of validating on-target effects, for example by investigating the inhibitor after a phosphatase gene knockout.

### 5.3. F1063-0967

Lastly, as mentioned previously Ren et al. screened a chemical library in silico to identify small molecule inhibitors of DUSP26 [33]. Initial homology modelling of the DUSP13B and DUSP26 structures generated a conformationally active pharmacophore model of DUSP26. Subsequently, a virtual screen of 130,000 compounds identified F1063-0967 as a potent inhibitor with an IC50 of 11.62 µM [33]. F1063-0967 increased apoptosis in IMR32 cells, in accord with previous experiments in NB cells where DUSP26 knockdown resulted in p53- and p38-mediated apoptosis [30]. Promisingly, as substrate selectivity is evidently the largest hurdle to overcome when developing DUSP26 inhibitors, the top eight compounds in their screen shared five different scaffolds, suggesting different DUSP26 binding mechanisms. Structure–activity relationship (SAR) studies on these scaffolds will likely yield more selective and potent inhibitors of DUSP26. Although it is unlikely the inhibitors discussed here will be used clinically, they are useful tools for understanding DUSP26 function and their scaffolds serve as good starting points for further SAR studies. The differences in structural topology and charge between the DUSP26 active site and its close relatives give hope that selective, orthosteric and allosteric inhibitors will be identified in the future [43].

## 6. Conclusions

DUSP26 has emerged as a phosphatase with a wide range of physiological substrates, both MAPK- and non-MAPK-related, and is implicated in several human diseases. An underlying theme of DUSP26 function is the apparent requirement of a binding partner to bring DUSP26 into close proximity to the substrate, to possibly increase DUSP26 activity through proximity or allostery. To date, this requirement has not been identified for other DUSPs. Further investigation into the residues that govern the DUSP26–binding partner interaction may reveal novel areas of the DUSP26 protein which could be targeted for chemical inhibition. There is emerging evidence of DUSP26′s role in malignancy. A large body of somewhat conflicting evidence implicates DUSP26 in the progression of NB; however, it is unclear whether DUSP26 is tumour-promoting or tumour-suppressing. In primary NB samples, *DUSP26* mRNA expression correlated with increased overall survival, whereas molecular studies demonstrate oncogene-like roles. Considering the great complexity of the tumour cell microenvironment in vivo, it is plausible that the molecular studies in vitro do not accurately represent the physiological function of DUSP26. In contrast, DUSP26 surfaces as a tumour suppressor in the majority of other cancers. A case in point is glioblastoma, where *DUSP26* mRNA expression is reduced in tumour specimens and inversely correlates with the histological grade of the tumour. Although overexpression of DUSP26 in glioblastoma cells suppresses cell proliferation, the physiological role of DUSP26 remains unclear. Finally, there have been some significant advances in targeting DUSP26 with small molecule inhibitors. Nevertheless, these inhibitors still have relatively poor selectivity and await further refinement. It will then be of great interest for these and future inhibitors to be tested for their therapeutic value in pre-clinical models.

## Figures and Tables

**Figure 1 ijms-22-00776-f001:**
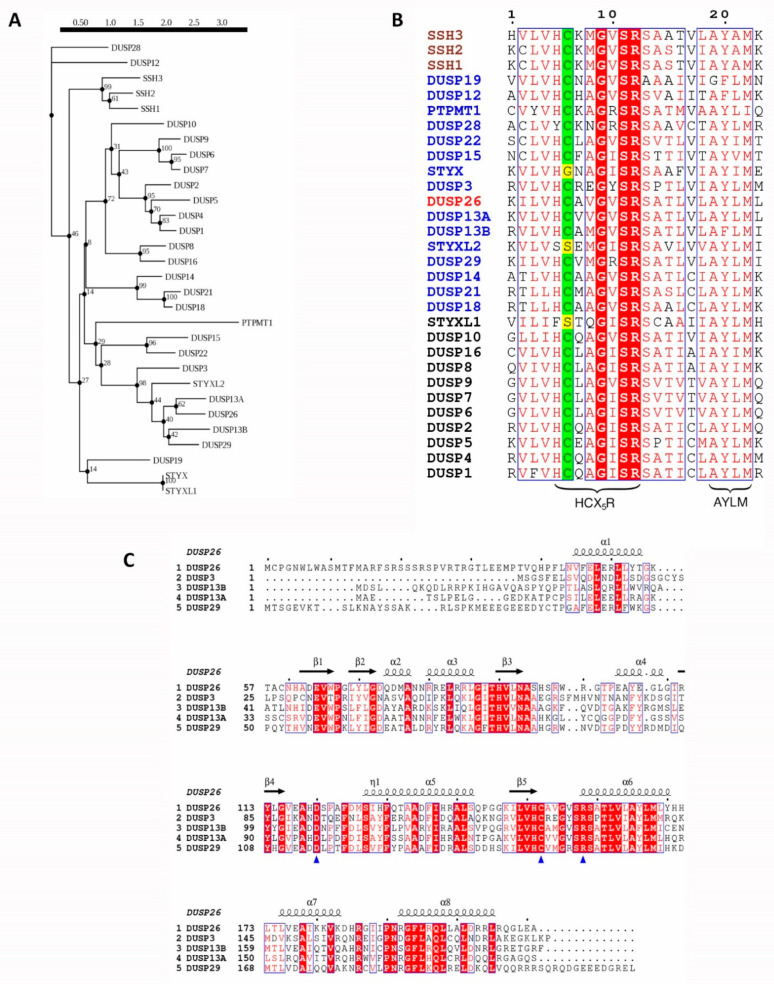
Evolutionary analysis of human DUSP26 and its homologous proteins. (**A**) Phylogenetic tree of human DUSP26 and its 29 closest homologous proteins. Sequences were identified using the blastp program in the UniProtKB/Swiss-Prot database. Following this, sequences were mapped onto a phylogenetic tree using the Smart Model Selection in PhyML with 100 bootstrap replications applied [37,38]. (**B**) Protein sequence alignment of the active site region of DUSP26 and 29 closest homologous protein sequences. Catalytic cysteine highlighted in green; pseudo-catalytic residue of pseudophosphatases in yellow; MKP, aDUSP and SSH names are in black, blue and brown, respectively, and the DUSP26 name is in red. (**C**) Protein sequence alignment of DUSP26 and its closest homologues, excluding the pseudo-phosphatase STYXL2. Blue triangles pinpoint the catalytic triad and depicted above are the α-helices and β-sheet positions and of the DUSP26 structure. For sequence alignments, identical residues are displayed in white inside red boxes; residues with 70% similarity based on physico-chemical properties are displayed in red inside blue boxes; alignment generated using the ESPript 3.0 tool [39].

**Figure 2 ijms-22-00776-f002:**
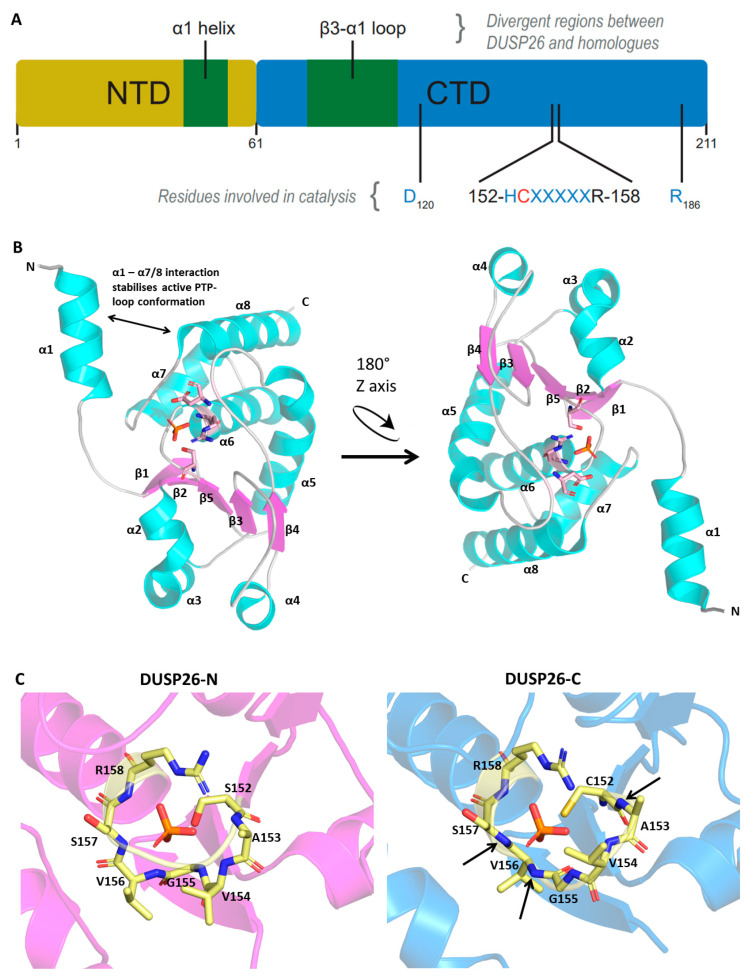
DUSP26 structure and comparison to homologous proteins. Protein Data Bank (PDB) file retrieved from the RCSB PDB Database and loaded into the Pymol software. (**A**) Schematic of protein domain structure depicting N- and C-terminal domains. (**B**) Crystal structure of DUSP26-N monomer presented with the catalytic triad surrounding the phosphate ion. (**C**) PTP-loop residues of DUSP26-N, DUSP26-C, DUSP3 and DUSP29 surrounding a phosphate (DUSP26-N and DUSP26C) or sulphate (DUSP3 and DUSP29) ion. The phosphate ion residing in the DUSP26-C PTP-loop has been superimposed from the DUSP26-N structure. Black arrows identify amide bonds in the DUSP26-C PTP-loop that are not pointing towards the phosphate ion centre. Structural alignment of DUSP26-N (magenta) with (**D**) DUSP3 (cyan) and (**E**) DUSP29 (green). Disparate structural regions are depicted by double-edged arrows and the catalytic Cys152 depicted as blue spheres. PDB codes: DUSP26-N, 5GTJ; DUSP26-C, 2E0T; DUSP3, 1VHR; DUSP29, 2Y96. Figures adapted from [43].

**Figure 3 ijms-22-00776-f003:**
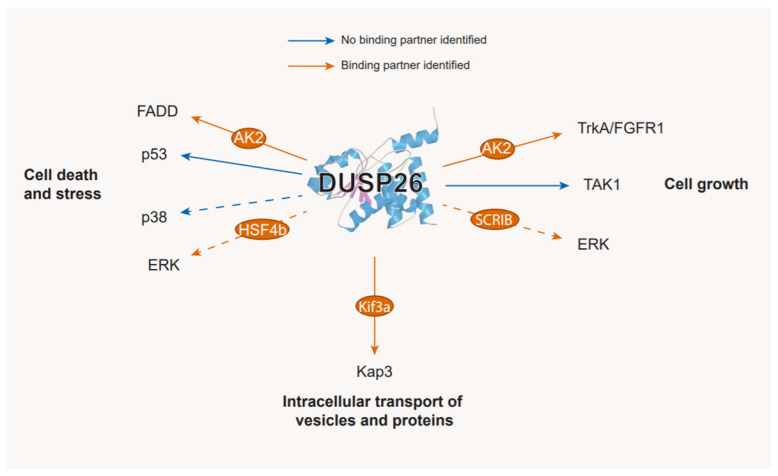
The substrates and binding partners of DUSP26. Displayed at the end of each arrow are DUSP26 targets that have been validated either in vitro or in vivo. Binding partners are depicted in orange circles along the arrows; substrates where no binding partner has been identified are depicted with blue arrows; substrates which require or are enhanced by a binding partner are depicted with orange arrows. Dashed arrows indicate substrates where the interaction has not been validated experimentally. See the text for further details.

**Figure 4 ijms-22-00776-f004:**
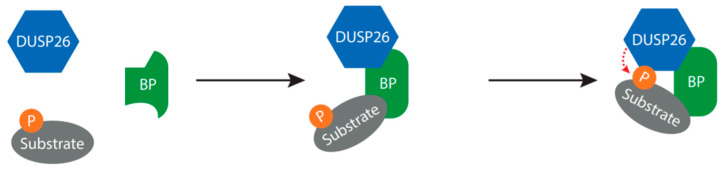
Proposed mechanism for DUSP26-mediated dephosphorylation. The binding partners discussed in this review are all able to bind both DUSP26 and the substrate, suggesting that the binding partner may function as a molecular bridge, required to bring DUSP26 into close proximity to the substrate and/or increase DUSP26 catalytic activity. Abbreviation: BP, binding partner; P, phosphate.

**Figure 5 ijms-22-00776-f005:**
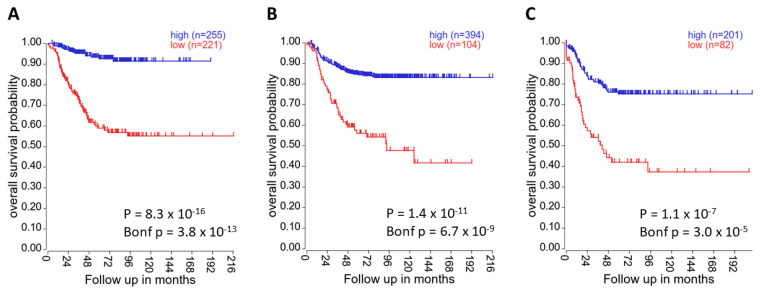
High *DUSP26* expression correlates with increased patient survival. In (**A**) the “Kocak-649-custom-ag44kcwolf” dataset, (**B**) the “SEQC-498-RPM-seqcnb1” dataset and (**C**) the “NRC-283-rma_sketch-(bc)-huex10t” dataset, *DUSP26* mRNA expression was significantly associated with overall patient survival (all *p* < 0.05). Kaplan–Meier survival curves were generated using the KaplanScan method presented on the R2 online platform [75].

**Table 1 ijms-22-00776-t001:** Summary of the atypical dual specificity phosphatases (DUSPs). Adapted from [8], see additional references in the text.

Gene Name	Synonyms	Chromosome Location	Molecular Weight * (kDa **)	Dephosphorylated Substrates
*DUSP3*	*VHZ; LDP3; MOSP; LDP-3; DUSP25*	1q23.2	20	ERK, JNK, p38, STAT5, FAK, EGFR, NPM1, NUCL, NBS1, HRNPC [9]
*DUSP11*	*PIR1*	2p13.1	44	5′-phosphorylated RNA CAP
*DUSP12*	*YVH1; DUSP1; GKAP*	1q23.3	38	Glucokinase, ASK1 [10], ERK, JNK, p38
*DUSP13A*	*BEDP; MDSP; TMDP; SKRP4; DUSP13B*	10q22.2	21	-
*DUSP13B*	*BEDP; MDSP; TMDP; SKRP4; DUSP13A*	10q22.2	22	JNK, p38, ERK
*DUSP14*	*MKP6; MKP-L*	17q12	22	ERK, JNK, Otulin [11], TAK1 [12], TAB1 [13]
*DUSP15*	*VHY; C20orf57*	20q11.21	32	-
*DUSP18*	*DSP18; DUSP20; LMWDSP20*	22q12.2	21	JNK, SHP2 [14]
*DUSP19*	*SKRP1; DUSP17; LMWDSP3; TS-DSP1*	2q32.1	24	JNK
*DUSP21*	*LMWDSP21*	Xp11.3	22	-
*DUSP22*	*VHX; JKAP; JSP1; MKPX; JSP-1; MKP-x; LMWDSP2; LMW-DSP2*	6p25.3	21	ERK, JNK, p38, STAT3, Erα, FAK [15], Lck [16]
*DUSP23*	*VHZ; LDP3; MOSP; LDP-3; DUSP25*	1q23.2	17	GCM1, p38, β-catenin
*DUSP26*	*LDP4; MKP8; NEAP; DSP-4; LDP-4; MKP-8; NATA1; SKRP3; DUSP24*	8p12	24	EGF, p38, FADD, TrkA [17], FGFR1 [17], TAK1 [18], KAP3 [19]
*DUSP28*	*VHP; DUSP26*	2q37.3	18	-
*DUSP29*	*DUPD1; FMDSP; DUSP27*	10q22.2	25	-
*EPM2A*	*EPM2; MELF*	6q24.3	37	GSK3β, Glucans
*PTPMT1*	*MOSP; PLIP; DUSP23; PNAS-129*	11p11.2	23	SDHA [20], PI(3,5)P2, PI5P, PGP [21]
*RNGTT*	*HCE; HCE1; hCAP; CAP1A*	6q15	69	5′-phosphorylated RNA Cap
*STYX*	*-*	14q22.1	25	-
*STYXL2*	*DUSP27*	1q24.1	130	-

* Predicted molecular weight retrieved from the Ensembl genome browser, ** Kilodalton.

**Table 2 ijms-22-00776-t002:** Tumour-suppressive and -promoting roles of DUSP26 in cancer.

Cancer	Relationship to DUSP26
Anaplastic thyroid cancer	Overexpressed in ATC tumours and cell lines. DUSP26 overexpression inhibited asinomycin-induced cell death [31].
T-cell lymphoblastic lymphoma	DUSP26 overexpressed in T-LBL tumour samples [57].
Medulloblastoma	*DUSP26* upregulated in medulloblastoma specimens [58].
	DUSP26 mRNA downregulated in medulloblastoma cell lines.
Neuroblastoma	DUSP26 overexpressed in NB cell lines and NB primary tissue. DUSP26 prevented Doxorubicin-induced cell death via p53 inhibition [59].
	In NB cell lines and intrarenal mouse model, DUSP26 shRNA or treatment with NSC-87877 inhibits cell proliferation [30].
	F1063-0967 induced apoptosis in IMR32 cells but not in HL7702 control cells [33].
	DUSP26 mRNA downregulated in NB cell lines [50].
Breast cancer	In SUM-52 cells, *DUSP26* amplified and DUSP26 shRNA decreased cell growth [60].
	In breast tumour samples, both loss and gain of *DUSP26* associated to patient overall survival [29].
	Reduced *DUSP26* expression in breast cancer tumours [61].
Colorectal cancer	HSF4b overexpressed in CRC and is inhibited by DUSP26 [56].
Pheochromocytoma	In PC12 cells, DUSP26 suppressed EGF-induced cell growth by downregulating AKT/PI3K signalling and EGFR transcription [17].
	In PC12 cells, DUSP26 suppressed NGF-induced neuronal differentiation by downregulating AKT/PI3K, TrkA and FGFR1 signalling [62].
Bladder cancer	Reduced *DUSP26* expression in bladder cancer specimens [63].
Lung cancer	DUSP26 hypermethylated in lung adenocarcinoma samples [51].
Sarcoma	*DUSP26* downregulated in human soft tissue sarcoma [64].
Pancreatic cancer	One patient with insulinoma had DUSP26 A160S mutation [65].
Glioblastoma	Reduced DUSP26 mRNA expression in glioblastoma specimens [19].
	*DUSP26* expression reduced in higher grade glioblastomas. High *DUSP26* expression decreased survival probability [28].
	In E98 cells, *DUSP26* overexpression decreased cell growth and motility [28].
	*DUSP26* part of a proneural glioblastoma signature which correlated with PATZ1, a GBM prognostic marker [66].
Ovarian cancer	In ovarian cancer cell lines, *DUSP26* deletion correlates with increased resistance to decitabine [50].
	Reduced DUSP26 mRNA expression in ovarian cancer cell lines [50].
	*DUSP26* part of a gene-signature that represses p53-p21 regulation in primary and recurrent murine embryonic carcinomas [67].

**Table 3 ijms-22-00776-t003:** Chemical inhibitors identified against DUSP26.

Inhibitor	Chemical Structure	IC50 (µM)	Additional Targets	Identification
F1063-0967	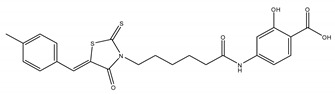	11.6 [33]	-	Screened DUSP26 pharmacophore model against 129,087 compounds in silico [33]
ethyl-3,4-dephostatin	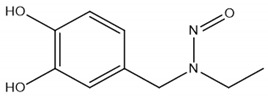	6.8 [32]	PTP1B, SHP-1 [102] DUSP14 [103] DUSP22 [104] PTPN2 [105]	In vitro phosphatase assay of ethyl-3,4-dephostatin against 13 PTPs [32]
NSC-87877	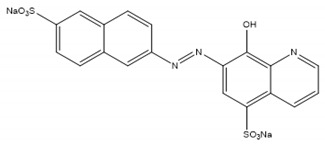	16.7 [34]	SHP1/2 [34,106] PTP1b, PTPN7, PTPRJ, PTPRC [106]	In vitro phosphatase assay of NSC-87877 against 15 PTPs [34]

## Data Availability

No new data were created or analyzed in this study. Data sharing is not applicable to this article.

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
