# Peer review of "A Review of DUSP26: Structure, Regulation and Relevance in Human Disease"

_ijms, 2021, doi:10.3390/ijms22020776_

Round 1
Reviewer 1 Report
DUSP26 is an atypical DUSPs that regulates multiple signaling pathway in cellular processes. Although it is unclear whether DUSP26 is tumour-promoting or tumour-suppressing, there has been an increasing interest in studying the inhibition of its activity. Unlike other DUSPs, DUSP26 requires a binding partner to bring DUSP26 into close proximity to the substrate for its function, which bring the complexity as well as opportunity for drug development. This review clearly describes the background, binding partners, structure, physiology roles and inhibitors of DUSP26. It will be of great interest to the reader of IJMS. The author should answer the following minor points before it is published.
- In line 47-49, “61 human DUSP genes have been identified, broadly divided into 6 classes: the MAP-Kinase Phosphatases (MKPs), slingshot phosphatases (SSHs), atypical DUSPs 48 (aDUSPs), PTENs, CDC14s and myotubularins [6].” In reference [6], DUSPs can be divided into six subgroups on the basis of sequence similarity that include slingshots, PRLs, Cdc14 phosphatases, PTENs, myotubularins, MKPs and atypical DUSPs. Please explain the discrepancy between this review and reference [6].
- In line 145-146, “the N-terminal α7-α8 loop stabilized the P-loop residues, G155-S157, creating an active PTP-loop conformation (Figure 2b).” In fact, the α7-α8 loop is located on the C-terminal.
- A recent study (IJMS 2020 Oct 12;21(20):7515.) indicated that a conserved active site configuration, the DPN-triloop interaction, played an important role in active site formation. This DPN triloop is conserved among the atypical DUSPs, including DUSP26. On the other hand, the active site, which comprises the D-loop, P-loop, and Q-loop, is highly conserved in classical PTPs. Since disruption of the DPN-triloop interaction can highly decrease the phosphatase activity, the results is similar to the DUSP26-C’s distorted P-loop. Author can consider to discuss in the review.
Author Response
Response to Reviewer 1 Comments
We appreciate the effort the reviewer has made in providing feedback on our manuscript and for the insightful comments. We have incorporated all of the changes to the manuscript suggested by the reviewers. Please see below our response to each point. These are also highlighted in the manuscript using tracked changes.
Point 1: In line 47-49, “61 human DUSP genes have been identified, broadly divided into 6 classes: the MAP-Kinase Phosphatases (MKPs), slingshot phosphatases (SSHs), atypical DUSPs 48 (aDUSPs), PTENs, CDC14s and myotubularins [6].” In reference [6], DUSPs can be divided into six subgroups on the basis of sequence similarity that include slingshots, PRLs, Cdc14 phosphatases, PTENs, myotubularins, MKPs and atypical DUSPs. Please explain the discrepancy between this review and reference [6].
Response 1: Thank you for highlighting this. In reference [6] the DUSP family is split into 6 groups in the text, where PTENs and myotubularins are part of the same group, but are divided into 7 groups in their Figure. The literature suggests there is a distinction between the PTENs and myotubularins and so they have been separated in our text, making it 7 groups.
Point 2: In line 145-146, “the N-terminal α7-α8 loop stabilized the P-loop residues, G155-S157, creating an active PTP-loop conformation (Figure 2b).” In fact, the α7-α8 loop is located on the C-terminal.
Response 2: Thank you for pointing out this typing error, it has been corrected to C-terminal.
Point 3: A recent study (IJMS 2020 Oct 12;21(20):7515.) indicated that a conserved active site configuration, the DPN-triloop interaction, played an important role in active site formation. This DPN triloop is conserved among the atypical DUSPs, including DUSP26. On the other hand, the active site, which comprises the D-loop, P-loop, and Q-loop, is highly conserved in classical PTPs. Since disruption of the DPN-triloop interaction can highly decrease the phosphatase activity, the results is similar to the DUSP26-C’s distorted P-loop. Author can consider to discuss in the review.
Response 3: Thank you for raising this paper with us. As you say, the distorted active site in the DUSP26-C structure may well disrupt key DPN-interactions in the active site. We agree that this is relevant to the review and have therefore speculated that these DPN interactions are likely disrupted in the DUSP26-C structure resulting in a decrease catalytic activity (line 146).
Additional Changes
Please note that we have taken the opportunity to make a number of further, largely minor text changes have been made and can be seen in tracked changes. These changes are for clarity purposes only and improve the accuracy and readability.
Reviewer 2 Report
In their review paper titled "A review of DUSP26: structure, regulation and relevance" by Thompson and Stocker the authors covered many aspects, ranging from 3D structure to the efforts made in developing DUSP26 specific inhibitors, regarding the atypical dual specificity tyrosine phosphatase DUSP26. On the whole I appreciated the manuscript. According to me there are just a couple of minor issues that deserve attention before that manuscript can be considered for publication in International Journal of Molecular Sciences.- Some typos: e.g. line 107 (Figure 1), for consistency with all the other figures, please refers as (Figure 1A);
- some sentences should be rephrased: e.g. a) lines 34-36 "Over 100 PTP supefamily family proteins......that is critical for dephosphorylation activity"; b) lines 105-108, it would be good merge the 2 sentences or alternatively make more clear the second one;
- line 42: regulation through splicing looks a post-transcriptional modification rather than a post-translational one;
- paragraph 3.1. "Transcriptional regulation": in lines 200-201 the authors stated "DUSP26 gene expression is downregulated epigentically by both histone acetylation and promoter methylation". In the next sentence they argued exactly the opposite. Please clarify the issue;
- Figure 3: it would be recommended replace the red and blue colors with solid arrows for the experimentally validated physical interactions and with dashed arrows for those not experimentally validated (e.g. p53), respectively;
- Table 2: differently from what it is indicated in the legend in the file I have got there are no highlighted green rows. Might it be the file I downloaded was corrupted. Please check;
- In the main text (lines 75-76) the authors highlighted that the selective PTPs inhibition currently remains an unmet need. However, later in the manuscript they discuss some recent successful achievements supporting a therapeutic potential of specifically targeting DUSP26 in neuroblastoma (paragraph 5.2. "NSC-87877"). Since, at least one of the authors recently published a paper in which he assessed the role of the nanodelivered oxivanadium on neuroblastoma cells my straightforward question is: is it plausible to speculate that the compounds reported in the paper might affect DUSP26 activity?
- Figure 5: please increase plots' font size because figures in brackets as well "high" and "low" once published, I guess, will be barely readable.
Author Response
Response to Reviewer 2 Comments
We appreciate the effort the reviewer has made in providing feedback on our manuscript and for the insightful comments. We have incorporated all of the changes to the manuscript suggested by the reviewers. Please see below our response to each point. These are also highlighted in the manuscript using tracked changes.
Point 1: Some typos: e.g. line 107 (Figure 1), for consistency with all the other figures, please refers as (Figure 1A);
Response 1: We have modified the main text so that it is consistent with the figure legend.
Point 2: some sentences should be rephrased: e.g. a) lines 34-36 "Over 100 PTP supefamily family proteins......that is critical for dephosphorylation activity"; b) lines 105-108, it would be good merge the 2 sentences or alternatively make more clear the second one;
Response 2: We agree and in a) the sentence has been re-written and in b) the two sentences have been merged into one as this is clearer to read.
Point 3: line 42: regulation through splicing looks a post-transcriptional modification rather than a post-translational one;
Response 3: We agree and have incorporated this suggestion.
Point 4: paragraph 3.1. "Transcriptional regulation": in lines 200-201 the authors stated "DUSP26 gene expression is downregulated epigentically by both histone acetylation and promoter methylation". In the next sentence they argued exactly the opposite. Please clarify the issue;
Response 4: Thank you for identifying this typing error. Demonstrated in reference [52], DUSP26 expression is downregulated by de-acetylation of histones, therefore, the text (line 226) has been corrected to “de-acetylation”.
Point 5: Figure 3: it would be recommended replace the red and blue colors with solid arrows for the experimentally validated physical interactions and with dashed arrows for those not experimentally validated (e.g. p53), respectively;
Response 5: We agree with this comment and dashed lines have been incorporated into Figure 3. The addition of dashed lines for experimentally validated physical interactions adds useful information to the figure. In reference [69], a physical interaction between DUSP26 and p53 has been identified by in vitro and so we have used a solid arrow to depict this. In accord with this, the validated DUSP26-p53 physical interaction has been incorporated to the main text (lines 337-338).
Point 6: Table 2: differently from what it is indicated in the legend in the file I have got there are no highlighted green rows. Might it be the file I downloaded was corrupted. Please check;
Response 6: Thank you for highlighting this but the rows in Table 2 appear green in our document. We have taken the opportunity though to change the red rows to blue, since blue and green remain more distinct for people who are colour blind.
Point 7: In the main text (lines 75-76) the authors highlighted that the selective PTPs inhibition currently remains an unmet need. However, later in the manuscript they discuss some recent successful achievements supporting a therapeutic potential of specifically targeting DUSP26 in neuroblastoma (paragraph 5.2. "NSC-87877"). Since, at least one of the authors recently published a paper in which he assessed the role of the nanodelivered oxivanadium on neuroblastoma cells my straightforward question is: is it plausible to speculate that the compounds reported in the paper might affect DUSP26 activity?
Response 7: In the main text it is stated that inhibitor generation against specific PTPs is notoriously difficult. Considering the achievements in inhibitor generation documented later on, we agree that this could confuse the reader. Therefore, we have modified the text to make the distinction clearer. Thank you for the comment on the potential effect of oxidovanadium compounds on DUSP26 activity. We agree that this is relevant to the review and have incorporated it into the main text (lines 494-499). As oxidovanadium compounds result in broad non-specific inhibition of PTPs, it is likely to disrupt DUSP26 activity and this could be partly driving the phenotypic changes observed in neuroblastoma cells.
Point 8: Figure 5: please increase plots' font size because figures in brackets as well "high" and "low" once published, I guess, will be barely readable.
Response 8: We have increased the plot font size of Figure 5 so that it is readable.
Additional Changes
Please note that we have taken the opportunity to make a number of further, largely minor text changes have been made and can be seen in tracked changes. These changes are for clarity purposes only and improve the accuracy and readability.